# Ovarian Tissue Cryopreservation for Fertility Preservation in Patients with Hemoglobin Disorders: A Comprehensive Review

**DOI:** 10.3390/jcm13133631

**Published:** 2024-06-21

**Authors:** Catherine Haering, Kathryn Coyne, Katherine Daunov, Samuel Anim, Mindy S. Christianson, Rebecca Flyckt

**Affiliations:** 1Division of Reproductive Endocrinology and Infertility, Case Western Reserve University School of Medicine, Cleveland, OH 44106, USA; cph43@case.edu; 2Department of Reproductive Endocrinology and Infertility, University Hospitals Ahuja, Beachwood, OH 44122, USA; 3Department of Hematology and Oncology, University Hospitals Seidman Cancer Center, Cleveland, OH 44106, USA; 4Department of Reproductive Endocrinology and Infertility, Cleveland Clinic Foundation Beachwood Family Health Center, Beachwood, OH 44122, USA

**Keywords:** ovarian tissue cryopreservation, hemoglobin disorders, sickle cell disease, beta thalassemia, fertility preservation

## Abstract

Hemoglobin diseases like sickle cell disease (SCD) and β-thalassemia (BT) present fertility challenges for affected patients. SCD and BT result from abnormal hemoglobin production or structure and pose numerous health concerns. Despite medical advancements improving the quality of life or even providing cures, SCD and BT pose unique fertility concerns for women. Young women with these disorders already contend with reduced ovarian reserve and a narrower fertile window, a situation that is compounded by the gonadotoxic effects of treatments like medications, transfusions, stem cell transplants, and gene therapy. While crucial for disease control, these interventions may lead to reproductive health issues, increasing infertility and early menopause risks. Ovarian tissue cryopreservation (OTC) offers potential for future motherhood to women with hemoglobin disorders facing infertility related to curative treatments. OTC involves surgically removing, preparing, and freezing ovarian tissue containing primordial follicles capable of producing mature oocytes, offering advantages over oocyte cryopreservation alone. However, the application of OTC for patients with hemoglobin disorders presents unique challenges, including special health risks, financial barriers, and access to care. This comprehensive literature review delves into the current state of ovarian tissue cryopreservation for fertility preservation in patients with hemoglobin disorders. Empowering patients with informed reproductive choices in the context of their hemoglobin disorders stands as the ultimate goal.

## 1. Introduction

Hemoglobin diseases are chronic genetic disorders with autosomal recessive inheritance that involve the abnormal production or structure of hemoglobin protein, with sickle cell disease (SCD) and β-thalassemia (BT) being the two most common disorders worldwide. SCD is a hemoglobinopathy characterized by a mutation in the hemoglobin protein, causing misfolding and leading to shortened red blood cell (RBC) lifespan and a hypercoagulable state. BT is a group of autosomal recessive disorders characterized by decreased or faulty synthesis of hemoglobin chains, leading to variable states of chronic anemia. While advancements in medical care have significantly improved the quality of life for individuals with SCD and BT, the disease poses unique challenges for women who desire to preserve their fertility. Young women with SCD already have a narrower fertile window from reduced blood flow to the ovaries and increased incidence of infarcts [1]. Patients with BT have increased fertility risks from frequent blood transfusions, leading to endocrine disruptions including decreased baseline anti-Müllerian hormone and hypogonadotropic hypogonadism [2]. The treatments used for hemoglobin disorders are also often gonadotoxic and can range from blood transfusions (i.e., iron overload effects) and chemotherapeutic hematopoietic stem cell transplantation and gene therapy to and long-term medication regimens (e.g., hydroxyurea) [3,4]. These therapeutic interventions, while essential for disease control, can have detrimental effects on reproductive health, potentially leading to premature ovarian failure or diminished ovarian reserve [5]. A 2023 study showed that while most patients with hemoglobinopathies wish to have children, few patients actually recalled receiving fertility counseling or testing [6]. Consequently, women with SCD and BT face an increased risk of infertility and early menopause, thereby compromising their ability to conceive and bear children in the future.

In recent years, ovarian tissue cryopreservation (OTC) has emerged as a promising option for women facing the potential loss of fertility due to hemoglobin disorders and their associated gonadotoxic treatments. The American Society for Reproductive Medicine officially lifted the experimental label on OTC procedures in December 2019 [7]. OTC involves the surgical removal, preparation and freezing of strips of ovarian tissue, which contains primordial follicles capable of producing mature oocytes via orthotopic or heterotopic transplantation of tissue when a patient desires conception [8]. This technique holds several advantages over traditional methods such as oocyte or embryo cryopreservation, including independence from ovarian stimulation, immediate preservation capacity, and potential restoration of ovarian endocrine function. 

However, while ovarian tissue cryopreservation holds immense promise, several challenges and considerations exist in its application, specifically for women with SCD and BT. These include specific risks associated with hemoglobin disorders, financial barriers, and access to specialty care. This comprehensive review article aims to explore the current state of knowledge regarding ovarian tissue cryopreservation for fertility preservation for this cohort. By examining the existing evidence, we hope to pave the way for further research and clinical advancements, ultimately increasing awareness around fertility preservation and access to comprehensive care centers for patients from all backgrounds.

## 2. Materials and Methods

In this comprehensive review, we utilized three major databases to find relevant peer-reviewed research articles and review papers: PubMed, NCBI, and ScienceDirect. The search was conducted for articles published from 1997 to 2023. We employed a combination of Medical Subject Headings (MeSH) search terms, including “ovarian tissue cryopreservation”, “sickle cell disease”, “beta-thalassemia”, “hemoglobin disorders”, and “fertility preservation”. The search strategy was designed to capture a broad range of studies and reviews related to the topic to ensure a comprehensive overview of the existing literature. 

We focused primarily on peer-reviewed primary research articles and review papers. The quality of the included studies was assessed using standardized criteria tailored to the type of study. For primary research articles, we evaluated methodological rigor, sample size, and the robustness of the reported outcomes. Review articles were assessed based on the comprehensiveness of the literature search, the quality of evidence, and the clarity of conclusions. Data from the selected studies were synthesized to provide an overview of the current evidence on the efficacy and feasibility of OTC for fertility preservation in patients with SCD and BT. We conducted a narrative review of the findings, highlighting key trends, common challenges, and gaps in the existing literature.

## 3. Results and Discussion

### 3.1. Physiology of Hemoglobin Disorders

Vaso-occlusive phenomena and hemolytic anemia are the major clinical effects of sickle cell disease and other hemoglobinopathies. Hemoglobin S (Hb S) results from the substitution of a valine for glutamic acid on the beta-globin chain, which produces a hemoglobin tetramer (alpha2/betaS2) that is poorly soluble when deoxygenated. Polymerization of the deoxygenated Hb S leads to vaso-occlusive events when the polymer distorts the red blood cell into the crescent or sickle shape [9]. In addition to polymerization, there are subsequent changes in red cell membrane structure and function that play a role in the pathophysiology of hemoglobinopathies. 

Homozygosity for Hb S or compound heterozygosity for Hb S and one of another HBB variant (including beta thalassemia) may cause the phenotype of SCD. This disease and others are characterized by life-long hemolytic anemia and intermittent vaso-occlusion, which results in recurrent painful episodes, ischemia-reperfusion injury, and end-organ damage, leading to life-long disabilities and shortened overall survival.

Beta-thalassemia is characterized by inherited mutations in genes that code for the beta chain of hemoglobin. The decreased synthesis of beta chains leads to variable severity chronic microcytic anemia, often treated with regular blood transfusions for symptomatic individuals. The severity of BT varies depending on how many of the four genes are mutated, ranging from mild, asymptomatic anemia to Hb Bart, which is incompatible with life and usually results in hydrops fetalis and early neonatal demise. In more severe disease forms, the loss of beta chains causes excess alpha hemoglobin chains to form insoluble polymers inside red blood cells, resulting in hemolysis and exacerbated anemia.

Chronic multi-organ dysfunction refers to the risks of stroke, cardiorespiratory disease, renal failure, and infection but also includes risk to the reproductive organs. SCD and BT have been reported to be associated with gonadal dysfunction in men and women, including both primary and secondary hypogonadism [10,11,12,13]. Similarly, the ovary is an end organ that can also experience damage. The literature on this topic is limited, as it relates to ovarian function in SCD and BT, but early studies have shown a delay in menarche and sexual development in females with these hemoglobinopathies [14,15]. Delayed menarche and sexual development also leads to a narrower fertility window for patients desiring children, further complicating their treatment and fertility goals [16,17]. 

It has been hypothesized that chronic transfusion and iron overload in patients with severe manifestations of hemoglobin disorders leads to gonadal dysfunction [2,18]. It has also been suggested that chronic anemia may result in vessel occlusion, infarction, and tissue hypoxia that results in ovarian dysfunction and premature ovarian insufficiency in patients with hemoglobinopathies [19]. Indeed, a case–control study reported that women of reproductive age with sickle cell disease may have a lower ovarian reserve at a younger age in comparison with patients with no hemoglobinopathy, though this may also be confounded by demographic factors such as race [1].

Sickle cell disease and beta-thalassemia are two of the most common genetic hemoglobinopathies in the world, and there have been significant advancements in detection and management of hemoglobinopathies [20,21]. Given the increased life expectancy from their teenage years to well into their fifth decade, patients should be offered care that not only ensures their survival but also supports their quality of life and reproductive potential [22,23]. 

### 3.2. Treatment of Hemoglobin Disorders

For female patients with hemoglobin disorders, there is little known about the effects of chronic medication use and curative treatments on fertility outcomes. Moreover, patients with hemoglobin disorders are already at increased risk for reduced fertility from diminished perfusion and increased incidence of infarcts at the ovaries [1]. Patients with BT commonly receive frequent blood transfusions to correct chronic anemia; one unit of transfused blood contains approximately 200–250 mg of iron, which is 2–4 times what is considered the normal range of blood iron in a healthy woman [24]. Frequent transfusions and elevated iron concentrations over time are shown to cause liver damage, cardiomyopathies, endocrine dysfunctions, and compromised function of the reproductive organs over time [13]. Children born with more severe forms of BT often receive regular blood transfusions starting at 2–6 months of age [25]. Although the mechanism remains unclear, transfusion-related hemochromatosis and suboptimal chelation therapy in childhood may lead to gonadal dysfunction, namely hypogonadotrophic hypogonadism via disruption of the hypothalamic–pituitary–ovarian axis [13]. For SCD and BT patients receiving chronic transfusions, one study found that 33% of BT patients had gonadal failure compared with 0% of SCD patients [26]. Proper chelation therapy is an essential part of treatment for BT and SCD starting in childhood. 

Long-term medications that reduce SCD symptoms include hydroxyurea, L-glutamine, and Crizanlizumab. While hydroxyurea is the oldest and most common treatment for SCD due to its ability to decrease the incidence of thrombosis and pain crises, hydroxyurea has been associated with diminished ovarian reserve and even miscarriage in women with SCD [17]. That said, hydoxyurea can allow SCD patients to lead healthier lives with fewer pain crises, and this may have fertility benefit. Ultimately, there are few concrete, high-quality clinical data on the effects of long-term hydroxyurea treatment in SCD and its effects on fertility [27]. While this remains an area of future research, patients should be counseled on the potential fertility risks related to long-term treatment of their hemoglobin disorder. 

The only known curative treatments for SCD and BT are hematopoietic stem cell transplant (HSCT) and gene therapy, which require preparatory gonadotoxic chemotherapy regimens. These chemotherapeutic agents include anti-thymocyte globulin, cyclophosphamide, fludarabine, and/or busulfan in high doses [28]. In some cases, total body radiation may be indicated. During radiation, testicles can be shielded, whereas ovaries cannot, placing patients with ovaries at much higher risk of infertility [28]. These chemotherapeutic agents are alkylating agents, meaning they form DNA cross-links and strand breaks, leading to cell death in maturing primordial follicles [29]. In a human xenograph model in vitro, a single dose of cyclophosphamide resulted in significant follicle death by apoptosis [29]. Ovarian failure and subsequent early menopause is a devastating and irreversible side effect of chemotherapy. Without pre-treatment fertility preservation, chemotherapy with HSCT has been shown to result in premature ovarian insufficiency in 89% of patients [30]. Because both chronic and curative treatment for hemoglobin disorders can be so damaging to the ovaries, patients desiring children should be counseled early in their disease course regarding options for fertility preservation.

### 3.3. OTC Mechanisms and Procedure

OTC involves the removal, preparation, and freezing of ovarian tissue containing primordial follicles, which harbor resting oocytes. This technique allows for the option for future re-implantation of cryopreserved tissue or potentially in vitro maturation of the follicles when the patient desires to conceive [8]. Ovarian tissue removal is most commonly performed laparoscopically and targets whole or partial ovarian cortical tissue to obtain the largest number of oocytes [31]. Although there are multiple methods described, one of the most common is to retrieve tissue, place in cooled media, and prepare the tissue in thin strips measuring approximately 10 mm × 5 mm × 5 mm sections prior to freezing (Figure 1) [32]. Various measures can be undertaken to avoid osmotic damage or ice crystal formation during the freezing and thawing of the ovarian tissue, including the use of various cryoprotective agents and vitrification versus slow freezing/thawing techniques [33]. OTC has not only shown success for women with SCD and BT but also for women with various types of cancer who have frozen tissue in preparation for similar gonadotoxic chemotherapies for treatment [34]. Over 200 live births after OTC and re-implantation have been recorded worldwide, with nearly all patients recovering ovarian function after re-implantation [33,35]. One 2023 study estimated that the spontaneous live birth rate after OTC for cancer or HSTC was 21% and 33% for OTC with IVF [36]. Even for cancer patients who underwent OTC prior to puberty, recent clinical evidence demonstrates successful live births after ovarian re-implantation after therapy [37]. For patients with SCD, limited data suggest greater barriers to live birth after OTC and re-implantation, but at least two live births have been reported for this group in the literature [34,38]. When the patient desires conception, cryopreserved ovarian tissue can be thawed and grafted back onto the ovarian medulla or placed within a peritoneal pocket for hopeful return of ovarian function and fertility [39].

### 3.4. Advantages of Ovarian Tissue Cryopreservation

OTC has become a standard of care in recent years for prepubertal women undergoing gonadotoxic treatment for SCD and BT. Prepubertal ovaries do not respond to ovarian hyperstimulation in the way that postpubertal ovaries respond, and thus, OTC is a viable alternative to preserve ovarian tissue that can be stored and subsequently used when patients are ready to discuss family planning after curative treatment [5]. There is also evidence that OTC performed at an earlier age avoids the hemo-occlusive and chelating effects of SCD and BT that would otherwise damage ovarian tissue over time [13]. In 2022, a calculator was developed to predict the effect of radiotherapy on fertility after treatment, which could help providers and patients assess the need for prophylactic fertility preservation [40]. It is also considered advantageous to perform OTC earlier in life, as women with hemoglobinopathies often face multiple routes to decreased ovarian reserve, including chronic inflammation, ovarian ischemia, and reperfusion injuries [26].

#### 3.4.1. Independence of Ovarian Stimulation

Unlike traditional assisted reproductive technologies, OTC does not require ovarian stimulation, making it a suitable option for patients with contraindications to hormonal stimulation, such as those with multiple comorbidities related to their hemoglobinopathy or who are undergoing acute complications and are too ill to undergo ovarian stimulation. Controlled ovarian hyperstimulation (COH) is typically an intense undertaking and is performed over 10–14 days of daily injections to induce multi-follicular development with the goal of collecting multiple mature oocytes for cryopreservation. For patients with SCD and BT, COH can pose numerous risks, including thrombosis from the increased estrogen state during stimulation and risk of triggering pain crises [5]. The promise of OTC and re-implantation of frozen–thawed ovarian tissue is the return of normal endocrine function after an SCD or BT curative therapy is established, with the production of viable eggs from the implanted tissue without the need for hormonal stimulation [41]. If IVF is desired, this process also allows for pre-implantation genetic testing for monogenic disorders (PGTM) to be performed, with the goal of selecting either unaffected embryos or an HLA-matched sibling embryo [42]. Interestingly, a 2013 publication indicated that only 24% of patients with an affected child with SCD were aware that PGTM was an option for limiting transmission of harmful alleles [43]. While patients may not be ready for children soon after OTC and curative therapy, education on third-party reproduction with gestational carriers and genetic testing may be helpful in informing patients on the full scope of reproductive options available.

#### 3.4.2. Immediate Preservation

Particularly for SCD, OTC can be performed at any time, including during an acute sickle cell crisis, without delay or interruption of treatment. A sickle cell crisis is a pain crisis that occurs in individuals with SCD when malformed RBCs block the small arteries that supply bones, causing severe pain for several days or even weeks [44]. If curative treatment such as HSCT is desired immediately, OTC is a viable option that obviates the need to go through hormonal stimulation (and avoids delaying curative treatments by weeks). However, the decision to undergo OTC would mean an additional laparoscopic surgery during a sickle cell crisis, which comes with its own perioperative risks. The care team and patient should work together to weigh the risks and benefits of the performing procedure during an acute sickle cell crisis.

#### 3.4.3. Potential Restoration of Endocrine Function

OTC enables the preservation of a large number of primordial follicles within the ovarian cortex, which allows for the potential restoration of endocrine function after re-implantation. The return of endogenous estrogen and oocyte production may increase fertility rates, delay the onset of menopause, and provide additional health benefits for patients with SCD and BT [26,39]. However, ovarian burnout is an important risk factor to consider when re-implanting ovarian tissue. Burnout is caused by factors such as hypoxia and damage during the freezing–thawing process and can reduce follicle survival as well as restoration of endocrine function [45]. A 2017 study showed that endocrine restoration, defined by cyclic menstrual cycles, ovarian follicle growth on ultrasound, or pregnancy, was observed in 64% of cases, with a clinical pregnancy rate after OTC at 57.5% [41]. A 2013 study showed even greater success, with 90% patients in Belgium, Denmark, and Spain who underwent OTC and re-implantation showing evidence of ovarian activity after four months [42]. However, timing is a critical consideration for when re-implantation should occur, especially for patients who underwent OTC prior to menarche. Even when pre-pubertal ovaries are re-implanted into patients, there is evidence of full restoration of endocrine function within 60 to 240 days, lasting up to seven years [37]. Because endocrine function may decline over time, providers should consider timing re-implantation closer to when the patient desires conception. The return of endocrine function is not only critical for pregnancy but also for mitigating the symptoms of hypoestrogenism and allowing the delay of early menopause.

### 3.5. Challenges and Disadvantages of Ovarian Tissue Cryopreservation in Patients with Hemoglobinopathies

#### 3.5.1. Limited Access

OTC requires specialized expertise in ovarian tissue cryopreservation procedures and workflows, limiting its accessibility for patients to centers with subspecialty expertise. With over 100,000 individuals diagnosed with SCD and 1200 individuals with transfusion-dependent BT in the United States, there is a significant need for adequate care to treat these patients’ unique medical needs and fertility concerns [46,47]. While BT remains a relatively rare disease in the United States, its disease burden is much more significant in Asian and Middle Eastern countries. On top of increased medical costs associated with SCD, around 40% of individuals are of African American descent, and 60% of individuals with SCD are publicly insured, have higher unemployment, and have lower lifetime earnings [3,48]. Racial and financial disparities in healthcare are exacerbated when complex and multi-specialty care is required. Even when awareness of fertility preservation options is achieved, and services are offered to patients with SCD and BT, the costs associated with OTC procedures and tissue storage are financial barriers that many are unable to overcome. Currently, there are 13 states with fertility preservation mandates requiring insurers to provide fertility preservation services when indicated; despite this, publicly funded health insurance is typically excluded from mandates, and it has also been demonstrated that African American patients have lower utilization of fertility services than other groups [4,49].

#### 3.5.2. Procedural Risks and Timing

Ovarian tissue removal is a laparoscopic procedure that poses additional risks for patients with hemoglobin disorders. Both surgical interventions and controlled ovarian hyperstimulation, if performed, can increase risk for adverse events including hypercoagulability, pain crises, and anemia in patients with SCD. The sickled erythrocyte results in intra- and extravascular hemolysis, shortened RBC lifespan, and a hypercoagulable state, particularly in smaller vessels [26]. This can cause sickle cell crisis, lasting days to weeks, with primary treatments aimed to manage symptoms, including hydration and analgesics [50]. COH, although not necessary for OTC and re-implantation, causes increased risk for sickle cell crisis, with a number of reported cases requiring hospital admission and RBC transfusion [51,52]. OTC, however, does have risks, as it requires a surgical procedure and careful peri-operative pain management; a sickle cell crisis was associated with an OTC procedure resulting in red cell exchange [53]. The medical team should take a multidisciplinary approach to OTC and COH procedures on patients with hemoglobin disorders due to their increased risk of coagulopathies and partner carefully with the treating hematology–oncology primary team as well as anesthesiologists and peri-operative care specialists.

#### 3.5.3. Risk of Re-Implantation Failure and Obstetric Complications

Successful transplantation and subsequent function of ovarian tissue after OTC is not guaranteed, and patients may require multiple re-implantation attempts to achieve pregnancy. While OTC shows significant promise for this patient population, it must be recognized that these hemoglobinopathies may adversely affect the ovaries from an early age before OTC can be performed as a preservative intervention. Sickle cell crises in SCD and hypogonadotropic hypogonadism in BT may damage ovarian tissue, reducing the efficacy of OTC at baseline. Although spontaneous pregnancy can occur after successful re-implantation, IVF may be required, which can incur additional and significant costs to the patient [26]. Finally, women with SCD and BT are at increased risk for complications related to pregnancy, including maternal mortality, hypertension, preeclampsia, preterm delivery, and miscarriage [26]. These risks are primarily associated with the hypercoagulable state seen in SCD and BT [54]. Management in pregnancy by a high-risk obstetrician with experience in hemoglobinopathies is preferred, as OTC alone may not be sufficient in severe cases.

## 4. Conclusions

Ovarian tissue cryopreservation is no longer considered an experimental procedure, although its long-term outcomes and success rates are still being investigated [7]. While curative therapies for SCD and BT have shown increasing success, their gonadotoxic nature limits the reproductive options and timeline for female patients with hemoglobinopathies desiring pregnancy. OTC has shown success in restoration of endocrine function and live birth in patients who underwent the procedure prior puberty before receiving curative therapy [37,52]. Variable restoration of endocrine function is a major advantage of OTC over other fertility options, such as oocyte or embryo cryopreservation, although some patients do still require IVF after OTC. Clinicians should be mindful that there are still increased risks associated with the tissue removal and transplantation procedures for patients with SCD and BT, including hypercoagulability, sickle cell crisis, and blood loss, worsening baseline anemia. Pregnancy also poses risks for both the patient and the fetus. Few data exist on success rates comparing pregnancy success for OTC re-implantation with and without IVF, and this is a potential area for future research. For women with significant health risks who wish to have biological children, use of a gestational carrier can be discussed as an option along with appropriate counseling from reproductive endocrinology, maternal–fetal medicine, hematology–oncology, and genetics specialists. Fertility preservation and the treatment of infertility in patients with hemoglobin disorders requires a multidisciplinary team and poses numerous financial and access-related barriers for disadvantaged or minority patients, leading to historically poorer treatment and fertility outcomes for this demographic. With OTC becoming an increasingly common procedure, advocating for accessibility and empowering patients and their families with informed reproductive choices remains the ultimate goal.

## Figures and Tables

**Figure 1 jcm-13-03631-f001:**
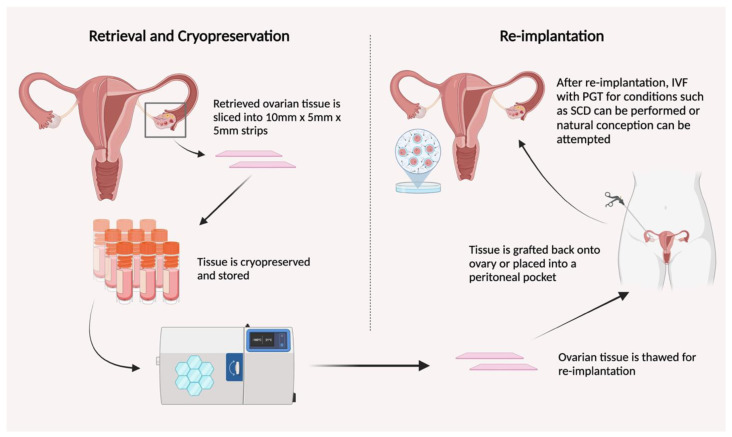
The process of retrieval and cryopreservation of ovarian tissue (**left**), followed by re-implantation of thawed tissue into the patient (**right**).

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
