# Peer review of "Ovarian Tissue Cryopreservation for Fertility Preservation in Patients with Hemoglobin Disorders: A Comprehensive Review"

_jcm, 2024, doi:10.3390/jcm13133631_

Round 1

Reviewer 1 Report

Comments and Suggestions for Authors

This narrative review is useful, well-structured and comprehensive. I have only minor comments and suggestions.

There are some studies that could usefully be incorporated into the review:

The effect of TBI on the ovarian reserve has been modelled, with a tool available to support decision making and patient counselling with respect to options for OTC. PMID 36399448.

A recent review gives updated information (including live birth rates) after female fertility preservation for cancer or haematopoietic stem cell transplantation.  PMID: 36421038

As noted in the manuscript, the prepubertal ovary is markedly different in terms of the dynamics of ovarian function after reimplantation, but there is recent emerging evidence that OTC is still a viable option:
"With respect to the prepubertal cohort, a spontaneous pregnancy resulting in live birth was reported in 2015, following autotransplantation of thawed ovarian tissue from a female who had undergone OTC prior to menarche. Additionally, it was observed that ovarian function resumed between 60 and 240 days post transplantation and persisted for up to 7 years. As the efficacy of autotransplanted ovarian tissue in maintaining endocrine function over the long-term is low, it is recommended to perform the procedure when the patient is preparing for conception [28]. After the favorable outcome, an additional 15 patients were documented. Of these, nine were diagnosed with a malignant ailment, whereas the remaining six were not. Conversely, five of the patients had not yet undergone menarche prior to OTC treatment, while eight had already undergone chemotherapy. Following ovarian tissue removal, all patients underwent gonadotoxic treatment. In 80% of the patients, ovarian function resumed, including 3 girls who were prepubertal at the time of OTC. Furthermore, of the 15 patients, 9 conceived at least once (60%) and 7 gave birth to at least one child (47%), including 2 who were not pubertal at the time of OTC [28,29,30,31,32]. Importantly, of the 15 patients, 9 conceived at least once (60%) and 7 delivered at least one child (47%), including 2 who were prepubertal at the time of OTC [28,29,30,31,32]." Quote from PMID: 37770817

In 3.4.2 the authors correctly point out the speed at which ovarian tissue can be obtained, but it is worth noting that an additional invasive laparoscopy procedure is involved, which could complicate the clinical decision making around the acute sickle cell crisis.

AMH was used as an ovarian reserve biomarker in the study that reported decreased fertility in women with SCD. But this study also reports: "Studies have shown significantly lower levels of serum AMH in black women in comparison to white women, as well as significantly greater age-related decline in AMH over time compared with white women [26, 28, 29]." So it maybe isn't clear at the moment that the observed reduced AMH was due to the condition (and not ethnicity/race).

Reviewer 2 Report

Comments and Suggestions for Authors

Although the subject is interesting, the study design is incomplete, and more information should be presented to analyze this review accurately. The entire text gives the impression of a simple presentation of other related articles without highlighting any special connection between them.

1.     Line 16 - Patients with ovaries sound weird. You can simply use the term women. 

2.   Line 35 - Hemoglobin diseases are chronic genetic disorders with autosomal recessive inheritance.

3.     In the introduction, in addition to describing sickle cell disease and how it negatively influences the ovarian reserve, you should also write about beta-thalassemia, the article referring to both hematological pathologies.

4.     By what mechanisms can the administration of iron affect the ovarian reserve and fertility (e.g., inhibits the proliferation of granulosa cells, etc.), and at what dose of iron (eg. 45 mg/day or higher or lower) and for what period, if these data are known and there are published studies. 

5.     Iron overload, in addition to ovarian damage, can influence fertility by decreasing endometrial receptivity and other effects that can be described in the introduction or results/discussion section. Can OTC alone solve the problem?

6.     In the title and abstract, you wrote that you did a comprehensive review, and in the methodology, you did a systematic review. I consider it a big error since there is a difference between the two types of studies that require a different search. Also, you can create a chart using the descriptive method of your search, with studies screened, excluded, and included according to the type of review. 

7.     The methodology should be more detailed. Also, in the abstract, you refer to sickle cell and beta-thalassemia, but beta-thalassemia is missing from MeSH terms. 

8.     As an introduction, in the results section, you only discussed sickle cell disease, omitting beta-thalassemia.

9.     By what mechanisms of gonadotoxicity do chemotherapeutic agents used for the treatment of hemoglobin disorders lead to infertility? Is this reversible ovarian failure or not? 

Round 2

Reviewer 2 Report

Comments and Suggestions for Authors

I congratulate the authors for the work done and the changes made according to the recommendations.

Kind regards